# Cellular Stress in Dry Eye Disease—Key Hub of the Vicious Circle

**DOI:** 10.3390/biology13090669

**Published:** 2024-08-28

**Authors:** Gysbert-Botho van Setten

**Affiliations:** 1St. Eriks Eye Hospital, 17164 Solna, Sweden; gysbert.van.setten@ki.se; 2Department of Clinical Neuroscience, Division of Eye and Vision, Lab of DOHF and Wound Healing, Karolinska Institutet, Eugeniavägen 12/Level 6, 17104 Solna, Sweden

**Keywords:** dry eye disease, stress, integrated stress response (ISR), adaption, translation, homeostasis, allostasis, heterostasis, resilience, coping, apoptosis, therapy, recovery

## Abstract

**Simple Summary:**

Dry eye disease includes a large variety of different ocular surface diseases. What all have in common is the resulting lubrication deficiency. Although the pathophysiology is multicausal, the common nominator is the threat of normal ocular surface balance and homeostasis. In homeostasis, the close link and interconnection of cellular basic mechanisms is essential for the functionality of the ocular surface. Any challenge threatening homeostasis implies cell stress. The ability of the ocular surfaces (conjunctiva and cornea) to cope and adapt to the resulting stress load decides over the ability to regain homeostasis or to drift to an altered stage of balance, recently identified as allostasis. This review outlines the importance of cellular stress as a key hub of the vicious circle and the importance of the integrated stress response system. Basic considerations for cellular recovery and the potential re-establishment of homeostasis are discussed.

**Abstract:**

Disturbance or insufficiency of the tear film challenges the regulatory systems of the ocular surfaces. The reaction of the surfaces includes temporary mechanisms engaged in the preservation of homeostasis. However, strong or persisting challenges can lead to the potential exhaustion of the coping capacity. This again activates the vicious circle with chronic inflammation and autocatalytic deterioration. Hence, the factors challenging the homeostasis should be addressed in time. Amongst them are a varying osmolarity, constant presence of small lesions at the epithelium, acidification, attrition with mechanical irritation, and onset of pain and discomfort. Each of them and, especially when occurring simultaneously, impose stress on the coping mechanisms and lead to a stress response. Many stressors can culminate, leading to an exhaustion of the coping capacity, outrunning normal resilience. Reaching the limits of stress tolerance leads to the manifestation of a lubrication deficiency as the disease we refer to as dry eye disease (DED). To postpone its manifestation, the avoidance or amelioration of stress factors is one key option. In DED, this is the target of lubrication therapy, substituting the missing tear film or its components. The latter options include the management of secondary sequelae such as the inflammation and activation of reparative cascades. Preventive measures include the enhancement in resilience, recovery velocity, and recovery potential. The capacity to handle the external load factors is the key issue. The aim is to guard homeostasis and to prevent intercellular stress responses from being launched, triggering and invigorating the vicious circle. Considering the dilemma of the surface to have to cope with increased time of exposure to stress, with simultaneously decreasing time for cellular recovery, it illustrates the importance of the vicious circle as a hub for ocular surface stress. The resulting imbalance triggers a continuous deterioration of the ocular surface condition. After an initial phase of the reaction and adaption of the ocular surface to the surrounding challenges, the normal coping capacity will be exhausted. This is the time when the integrated stress response (ISR), a protector for cellular survival, will inevitably be activated, and cellular changes such as altered translation and ribosome pausing are initiated. Once activated, this will slow down any recovery, in a phase where apoptosis is imminent. Premature senescence of cells may also occur. The process of prematurization due to permanent stress exposures contributes to the risk for constant deterioration. The illustrated flow of events in the development of DED outlines that the ability to cope, and to recover, has limited resources in the cells at the ocular surface. The reduction in and amelioration of stress hence should be one of the key targets of therapy and begin early. Here, lubrication optimization as well as causal treatment such as the correction of anatomical anomalies (leading to anatomical dry eye) should be a prime intent of any therapy. The features of cellular stress as a key hub for the vicious circle will be outlined and discussed.

## 1. Introduction

Ocular surfaces are, like any surface of the body, continuously challenged by environmental factors imposing a stress load on their basic component, the cell [1]. However, the ocular surfaces with the cornea and conjunctiva are unique with their immediate importance in the ability of an individual to interact with the environment. Challenging the ocular surfaces is equal to challenging vision as such. At the ocular surface, the tear film as a liquid medium protects the cells of the conjunctiva and the cornea against stress factors such as desiccation, mechanical friction, and attrition [2], functioning as a buffer [3]. The normal composition [4] and quantity of tear film warrant the sufficiency of their protective capability. Complementary to this, a plethora of protection mechanisms exist at the surfaces, such as the mucins [5,6]. The defense abilities of tear film and surfaces together initially attenuate any external stimulation and contribute to the cellular resilience. A certain level of constant challenges and nudging is, however, a physiological necessity and their silencing or even elimination is neither desirable nor possible [7]. If, on the other hand, the intensity of consistency of challenges exceeds the limits, excessive stress might occur. The resulting cellular changes in such excessive external stress become, with the current possibilities, initially clinically very subtle and mostly invisible [8]. This, in part, as short time stress excess, is usually managed within the normal scope of cellular resilience. The task is to define conditions in which ocular surface stress becomes pathological and threatens ocular surface integrity and functionality.

## 2. The Power of Surface Stress: The Pathophysiological Footprint and Impact

There is a significant difference in the response of cells to acute stress and chronic, persistent, or recurring stress [7]. The main difference is the cells’ inherent ability to cope with subcritical events. As long as it is not life-threatening directly to the cell, the cell reacts to acute stress exposure without major change in cell physiology. Spontaneous, temporary, and occasional stress on epithelial cells does hence cause usually only acute, transient endoplasmatic reticulum (ER) stress, which can, however, also decrease general protein synthesis [9]. Temporarily increased mRNA translations are followed by the normalization of protein synthesis and reconstitution of the cell to a normal functionality. If external stressors persist, continue, intensify, or recur frequently, this triggers a transition to a chronic stress response. This has been recently discussed with the introduction of the model of allostasis based on the model that DED is the result of a chronic stress condition. Overall, in cell physiology does the ability of cells to adapt to stress determine their ability to survive in a challenging environment [10]. In general, cellular response to stress ranges from the activation of survival pathways to the initiation of cell death that eventually eliminates damaged cells.

Although various coping mechanisms exist [1], the nature, intensity, and continuity of stress exposure determine the fate of a cell and the tissue it is part of. For example, with minor stress impact and with lower allostatic load, such as in moderate dry eye conditions, i.e., exposure to cold and wind [11], initially only a more limited stress response is launched. However, simultaneously, an enhanced level of cellular vigilance is triggered. As a result, the predisposition to a more destructive stress response could result from exposure to further stress, with the enhanced elimination of damaged cells [10]. Natural protection mechanisms in epithelia, such as the rapid cell turnover at surfaces such as cell shedding, contribute to the elimination of even potentially damaged cells at the layers close to the surface. The multilayered structure of the human corneal and conjunctival epithelium with five to seven cell layers [12] facilitates this process. In normal physiology, cell regeneration [13,14] and shedding [15,16,17] are essential, with the shearing forces of the lids being suggested to increase exfoliation and alter epithelial migration and turnover at the cornea [13]. The multilayering of epithelia hence contributes to the resilience of the surface, although numerical data are limited [18]. The observation that in DED the corneal epithelium was found to be thinner [19] could be of pathophysiological relevance as the associated decrease in resilience could result in increased sensitivity to ocular surface stress. Accordingly, in the model of attrition [20], the observed thinning of the corneal epithelium in dry eye disease results in increased sensitivity to shearing forces caused by lubrication insufficiencies.

In the current model of the multicausal pathogenesis of dry eye as a disease [21] each component has as a common feature an identity as stressor of homeostasis. In the current model of the vicious circle [22], one could consider each component as stressor that adds its share to the sum of all stressors summarized forming the allostatic load [1]. Considering this, the resulting total ocular surface stress experienced by the cells at the ocular surfaces hence becomes a major driving force, the key hub of the vicious circle. As long as the resilience of the ocular surfaces, i.e., their coping potential, exceeds the total ocular surface stress, the balance between the tear film and the ocular surface is preserved, and homeostasis is maintained. In this condition, one could consider the vicious circle [22,23] as silenced or dormant.

Challenges of the homeostasis such as a varying osmolarity, constant presence of small lesions at the epithelium, acidification, attrition with mechanical irritation, and onset of pain and discomfort all lead to a stress response. Insufficient lubrication, as the core event of dry eye disease, could enhance the susceptibility of the ocular surface to stress. With an increase in stress load and decrease in resilience in the system, the balance between the tear film and within the ocular surfaces becomes increasingly challenged, and, in the end, is lost. In this case, the vicious circle has been activated, threatening homeostasis and its potential reconstitution. It is probably at this stage when specific substances become detectable in tears, often referred to as “biomarkers”. The pathophysiological specificities of these markers, however, vary and remain to be clarified. Possibly, their designation as an “indicator” would be, for the time being, more justified until further data support their identity as a biomarker. The nature, i.e., the characteristics of stressors, determines the pathophysiological impact on the progress of DED. The definition of stress is hence very important.

## 3. Cellular Stress—Definition and Effects

Adaption to environmental challenges and changes is a prerequisite for cellular survival. Extracellular challenges incorporate a stress load which often is summarized as stress. The term stress has been modified and reevaluated over time [24], leading to the current understanding that stress is the challenge on a cell forcing it to adapt to the conditions around it and still able to maintain it’s homeostasis. The multicausal pathophysiology of DED reflects the plethora of different stressors, i.e., challenges to the ocular surface homeostasis. In DED the most well-known stressors are: tear film volume deficiencies, hyperosmolarity, as well as changes in tear film composition such as increase in metalloproteinase 9 (MMP-9) [25,26,27,28,29]. Other stressors are exposure to pollutants, ultraviolet (UV) radiation, ozone, and preservatives [30,31,32,33,34,35,36,37]. Additionally is oxidative stress known to be another important factor [38]. Amongst the factors listed, however, hyperosmolarity is this far the most commonly identified. It is currently considered as a major driving force in the vicious circle and a main stressor of dry eye disease [39,40]. Only recently different hyperosmolarity related effects such as alteration of cytoskeleton, cell cycle slowdown, adaptation mechanisms, apoptosis, and inflammatory response have being reviewed [41]. All of these specific stressors together propel the vicious circle. Additionally it may also be the sequence in which stressors align, activating linked components of the vicious circle at the ocular surface. The cellular stress as such is most likely its key hub. The intensity and persistency of stressors then determine how long the ocular surface does manage to preserve its functionality and is able to cope. Temporary challenges can be matched and controlled by a self regulating acute temporary inflammatory response system (earlier proposed as acute leucocytic response system, ALIRS) [42] with its necessary, time limited inflammatory reaction [43,44].

At this stage, with still preserved homeostasis, stress as external challenge on the cell can still be considered part of normal physiology. The ability of cells, such as epithelium of the conjunctiva and the cornea to cope with these challenges is essential for the maintenance of cellular viability and reactivity. The ability of cells to comfortably survive and strive in their environment reflects the balance between available normal stressors, i.e., normal allostatic load and the cells’ ability to deal with them. This is the essence of homeostasis. It is evident, that homeostasis depends on the established balance between the surface and lubricating fluid, i.e., the tear film. Within this system, there is a need for certain buffer function, that is an excess of capacity to allow sufficient lubrication even under slightly sub-optimal conditions. If either one or the other of the two components cannot cope with the demands of the other, homeostasis will be challenged, and eventually lost [1]. Eukaryotic cells respond to stress through adaptive programs [45]. The details of this adaption are complicate and not totally clarified. They include, however, the reversible shutdown of key cellular processes, the formation of stress granules, and a global increase in ubiquitination [46]. Any anatomical change of the surface profile such as microcysts, corneal scars, map and fingerprint dystrophy can cause an imbalance between lubrication capacity and anatomical needs [47]. At the ocular surface, also iatrogenic specific stressors are found in the micro anatomy of the surfaces, such as after laser surgery with contour changes at the flap edge; often in the presence of a normal tear film [47]. Similar conditions are found postoperatively in other surgery where the contour anomalies created at the surface supersede the capacity of the fluids to cover uneven surfaces, just as shown in the example of the epitheliopathy of the bleb [48]. In essence, it is the imbalance between micro anatomical requirement and lubrication capacity that brings the ocular surface-tear film system at the edge of its balance by exceeding the coping capacity [1,49]. The resulting enhanced friction, exceeding normal levels [50], results in enhanced mechanical transduction (attrition). This, as a major stress factor, triggers pre-inflammatory events [20]. Possibly it is important to differ between localized or general lubrication deficiencies, that is, local or total desiccation. This has been recently discussed with the model of the localized, anatomical dry eye [24]. Independent from this, is the clinical identification of an imbalance in the lubrication capacity and demand is inherently very difficult as no adequate specific tests do exist. Only after the ocular surface health and integrity has been challenged sufficiently, epithelial staining will become visible at cornea [51], and on the conjunctiva [52]. In order to facilitate staging a grading for the conjunctival surface alterations has most recently been suggested, the Sandbank Epitheliopathy [53]

As sensations may precede visible clinical symptoms [54,55], ignoring subjective symptoms and leaving without therapy can allow further increase or persistence of the allostatic load, eventually exceeding the copeable stress intensity and giving way to the development of detectable of anatomical changes. The basic clinical observations such as early breakup times, and disturbance of break up patterns hence have to be considered signs of functional lubrication insufficiencies known to occur in dry eye disease [56,57,58]. For the conjunctiva, enhanced ocular surface stress with the complex dysregulation of ocular inflammation may engage the conjunctiva-associated lymphoid tissue CALT [59,60,61,62] and result in a so-called circulus vitiosus with faulty differentiation of conjunctival cells [63].

In the vicious circle of DED as illustrated in DEWSII, hyperosmolarity plays a major role [21] and constitutes a major stress factor. However, it has been suggested that not necessarily the absolute value of osmolarity as such, rather the level of hyperosmolarity together with the magnitude of the osmotic diurnal variation. i.e., osmokinetics, strongly affect the stress impact [49,64]. In this model the desiccation stress caused by the diurnal variation of osmolarity could be amplified by an elevated average osmolarity level (Figure 1). This could cause an increase in desiccation stress.

Further rapid variations over a shorter time could result in an osmolarity jojo [2], depriving the cell from sufficient time for recovery in spite of the presence of mechanisms of cell recovery from osmotic stress [66]. Increased stress intensity inhibits osmoadaptation and activates pro-inflammatory gene expression (Figure 2).

For recovery, stress-free intervals would increase resilience and the ability of the ocular surface to cope with the next exposure to stress. The assumption that in adequately hydrated individuals, osmolarity and tears would be driven down to a basal level close to that of the plasma during overnight sleep, due to the prevention of tear evaporation as the eyes are closed [70], would support the idea that such periods of recovery would exist. However, tear production is considered to decrease during the night hours [71] with a simultaneous decrease in pH due to the continuous cell metabolism. Additionally, own investigations found elevated levels of osmolarity in the morning hours. This observation matches the report of aggravated symptoms experienced by patients suffering from DED directly after wakening [72]. Even enzymatic activities such as metalloproteinases have been found to be highest in the morning hours. It is hence more likely that the ocular surface stress continues even during the night hours, and that there is no significant relaxation or relief from, amongst others, osmotic stress, interrupting the vicious circle temporarily. Hence, there is most likely no nocturnal cessation of ocular surface stress, potentially promoting cellular recovery and enhancing the resilience of the ocular surface for the next day to come. The frequent occurrence of an incomplete closure of the eye lid (nocturnal lagophtalmos) [73] further enhances the dessication process and contributes to the rise in osmolarity in the remaining tears, over the entire surface. Especially here, with a reduced volume of the mare lacimale, eye movements during sleep may gain increased importance as they are needed to blend the fluid and its components. Such movements, on the other hand, are heavily influenced by the sleep characteristics and with this the dream activity [74]. Here, again the association with DED and depression [75] could further aggravate the viscous circle as antipressants can reduce the REM (rapid eye movement) phase during sleep [76,77] and decrease tear film production [78]. The nocturnal accumulation of neutrophils [79] with the associated presence of inflammatory mediators in the reduced volume of tear film, further support the concept of uninterrupted continuation of the vicious circle even at night.

Both, the intensity and the persistence of stress, does significantly influence cell physiology and changes basic cellular mechanisms [80]. In cells the stress response is coded by specific systems encoding, amongst others, the ubiquitin system [81]. This system is in turn challenged by the endoplasmatic reticulum (ER) stress [82,83]. In order to survive chronic stress cells need to adapt. Such adaption alters intercellular mechanisms guided by proline-rich receptor-like protein kinase (PERK). Adaptive, stress induced transcriptional reprogramming is closely coordinated with these events [9].

The accumulation of unfolded and misfolded proteins during stress is inherently toxic to cells and has to be quickly and efficiently handled. This is of particular importance during proteotoxic stress [82]. Eventually, the stress related accumulation of misfolded proteins can be a permanent burden and even risk cellular survival. Cellular cleanup mechanisms are therefore an essential part of physiology and include the evolutionarily conserved ER-associated degradation (ERAD) pathway [84], giving the cells a chance to recover from external stress. It is here, where the ubiquitin system and ubiquitin-like proteins play a key role [85]. A number of studies have shown that various proteotoxic stress conditions can cause functional impairment of the UPS (ubiquitin–proteasome system), resulting in cellular dysfunction and apoptosis [82]. Functionality of the UPS in turn is linked to the ER in that ER stress compromises UPS functionality [83].

Hyper-osmolarity as stress factor, not only influences cell volume [66] but also intracellular signaling [86,87]. The sensation to osmolarity changes can be conveyed via various primary osmo-sensors such as intracellular solute sensors, membrane-based osmo-sensors, and cytoskeleton associated osmo-sensors [87]. The osmo-sensory signal transduction network is closely linked to essential signaling mechanisms controlling critical physiological functions, such as immunity, apoptosis, proliferation, and differentiation [86]. Hence, triggering this system results in intracellular stress, requiring proper ER function. Impaired ER function in corneal epithelial cells, can lead to proinflammatory signaling and apoptosis. On the other hand, alleviation of ER stress could alone also enhance viability [88].

Permanent stress, such as the consistency of hyperosmolarity plays a decisive role for dry eye pathophysiology. On the other hand could the moderate, slower variation of osmolarity over the day qualify as short time stress without the induction of specific genes for the integrated stress response (ISR). This as the normal day to day variations within physiological limits are most likely part of a normal stress environment [89] and do, most likely, have physiological functions in the regulation of translation.

Chronic ER stress, on the other hand, induces a chronic ISR which could not only lead to a PERK kinase-driven translation initiation reprogramming but also, at the end, eventually lead to apoptosis and cell death [9]. However, this is not always mandatory as chronic severe stress can also push the cells to enter a transient hibernation-like state in anticipation of recovery [90]. The ubiquitin system with its specific codes plays a decisive role coordinating various cellular stress-associated responses, and support essential tasks such as adaptation, homeostasis recovery, and survival [81]. In general, it appears that the more and the longer a cell is exposed to stress, the longer its recovery will take. For some proteins the synthesis is restored to normal not earlier than 12 h after cessation of stress [90]. In the model of a vicious circle [22,23], hyperosmolarity with its continuing variations over the day serves as major stress factor. Influencing cellular environment apparently without cessation it could potentially contribute to the autocatalytic acceleration of severe dry eye pathophysiology. In the vicious circle, once activated, stress exposure increases whereas simultaneously time for cellular recovery decreases (Figure 3). 

## 4. Stress Management at the Ocular Surface and Recovery

The identification of core processes activated by stress also offers therapeutic opportunities. An intriguing option to enhance epithelial cell viability is hence the application of specific substances alleviating the ER stress. This has been recently shown for tauroursodeoxycholic acid (TUDCA), a chemical chaperone known to protect against ER stress [88].

The regulatory systems of the cell in response to stress includes, as one of the first steps, the modulation of key gene transcription and changes in the translation of proteins. This process, also referred to as adaptive translational pausing is a hallmark of the cellular response [90]. If this pausing lasts only for a short while, it is an essential part of the temporary compensatory mechanisms included in the envelope of homeostasis. Continued stress with the potential exhaustion of the buffering systems can lead to the entry in the vicious circle with chronic inflammation. This outlines the importance of stress management of cells as there is always an imminent link to inflammation [91]. Once triggered, inflammation can initiate an adaptive cell response driving inflammation [32] and activating MMPS in the epithelium of the cornea [92]. Tear deficient conditions do not have the same capacity as normal tears to neutralize MMP activity [93]. Additionally may inflammation as such, stress triggered or not, lead to altered production and profile of mucins at the surfaces [94]. The reduced production or quality of mucins at the surface [95,96,97] in DED [5,98] could increase attrition and contribute to further stress and inflammation. Once set in motion, the vicious circle of DED makes it increasingly difficult for the surface to recover with own resources.

## 5. Recovery from Stress and Stress-Related Changes

The nature of cellular changes at the ocular surface resulting from stress, determine the ability for and velocity to recover. As long as Allostasis with permanent changes of the ocular surface [99] has not been established, stabilization, and, eventually even recovery is still possible. For this recovery, aside from breaking the vicious circle as a first step, ubiquitination appears to essential [46]. The detailed processes of cellular recovery at the level of gene regulation is, on the other hand, so far unknown. Considering the timeframe, available results suggest that the stress duration dictates the availability of adaption programs needed for the promotion for return to homeostasis. Basically, it seems that the stress adaptation mechanisms allowing the cells to react to environmental load factors also determine the kinetics of recovery from stress [90]. Cellular reaction, for example to osmotic stress include immediate effects which are easily and swiftly reversible. If stress continues over longer time, adaptive processes within the cells occur [100,101], leading to longer time for recovery. Further data are needed to predict the effects of osmolarity changes [102]. Noteworthy is, that the stress response to osmolarity might differ when compared to other stressors [103]. In the worst scenario, if the tolerance threshold of cells in response to osmotic stress has been exceeded, programmed cell death might have been irrevocably initiated [87] and recovery has become impossible.

The level of cellular alteration in response to stress determines the mechanisms of recovery. Here the engagement of the adaptive response with the integrated stress response (ISR) [104] plays apparently a decisive role. The load factor of stress seems to be determined by the intensity of endoplasmic reticulum stress and its potential engagement of the ISR. (Figure 4). 

It is the potential ISR activation as a response to stress [90] that decides over the velocity of recovery. This as the reversal of the global translation initiation inhibition, ribosome turn on, ATG reactivation, handling of mitochondrial fragmentation and re-initiation of a normal cell cycle takes time. ISR activation is, on the other hand, the key event in the basic pathophysiological mechanism of chronic stress. The overlap of persistent ISR activation due to prevailing stress with other factors such as senescence or additional external load factors can lead to the impossibility of full recovery, and at the end to apoptosis with cellular death. In general does the duration of severe stress apparently dictate the threshold of gene regulation during recovery [90]. The duration of stress hence has a major impact of its own in addition to the effect of stress intensity. Both decide about the ability and velocity of recovery (Figure 5).

The therapeutic induction of cellular recovery from stress has to be conducted very carefully as cellular mechanisms have characteristics of its own. In situations when homeostasis has been lost and allostasis has taken place due to chronic stress, rearranged mechanisms well established might perceive a sudden deficit to stress exposure as an alternate stress. This “therapeutic” stress inversion can exceed the cells’ abilities to cope and survive. The identification of stress factors in the pathophysiology of DED should therefore imply a careful consideration how the stress factors are managed also therapeutically. Here indicators, preferentially biomarkers, may provide decisive information. Their presence alone, however, measured in concentration of activity [105], only reflect on the general condition of the ocular surface, and with this some insight on the surface cell’s ability to deal with the stress imposed. On the other hand, they could also provide a platform to judge over the cells ability to survive changes. Therefore, considering therapy, not the sudden elimination of stress, but its successful moderate but continued decrease of stressors could be the optimal way to go. If therapeutically a sudden withdrawal occurs, a “stress reversal” could be induced. The resulting sudden deficit of stress in an stress adapted system could lead to system overload, triggering inflammation, accelerating apoptosis and cell death. It could be here that immunomodulators such as corticosteroids [106,107], cyclosporine and others [108,109,110,111,112] play a key role in therapy of dry eye disease. With their capacity to ameliorate the inflammatory cascades, the vicious circle could be attenuated, and slowed down. With this, cells of the ocular surface could be put into position to adapt to stress reduction and avoid stress rebound effects. This would make potentially challenging therapy such as the use of, for example, hypotonic eyedrops less dramatic. Depending on the intensity of changes implemented in the cell metabolism during stress adaption, slow recovery might be advantageous and preferable, potentially even the only option. A slower, softer reconstitution of the ocular surface could be achieved by natural substances promoting cell recovery. For example, was recovery of corneas exposed to hyperosmotic stress significantly improved by treatment with 0.1% sodium hyaluronate [113]. This supports the increasing evidence that such substances as hyaluronic acid [114,115,116] should be considered as essential part of both initial and long term treatment in dry eye disease [117,118,119,120,121].

## 6. Conclusions and Outlook

Cellular stress has been identified as one of the crucial hubs of ocular surface disease when homeostasis is threatened and an altered stage of balance, defined as allostasis becomes imminent. Cellular stress itself is the result of all those factors challenging normal cellular physiology and leading to common pathways engaging the important switch of AP-1 (activating protein-1) transcription factor complex [122] and/or mitogen-activated protein kinase (MAPK) [123]. These pathways could be key nominators of the different compartments of the vicious circle [23]. As soon as ISR has been initiated, any recovery to homeostasis will take longer time- if it is still possible at all. It is potentially here, that the effects of immunomodulator treatment such as Cyclosporine A and steroids are located and have a common effect to ameliorate the dynamics of the vicious circle in DED. The effects of cell adaption to external stressors could also affect therapy, as the reversal of intracellular mechanisms resulting from adaption has to respect the cells sensitivity to the kinetics of changes in its proximity. Forced treatment can easily result in additional stress and possibly trigger adverse effects prior to improvement. Early and careful treatment of DED is hence advisable. To intense therapy, leading itself to intense stress exposure, can drive cells towards apoptosis [124].

## Figures and Tables

**Figure 1 biology-13-00669-f001:**
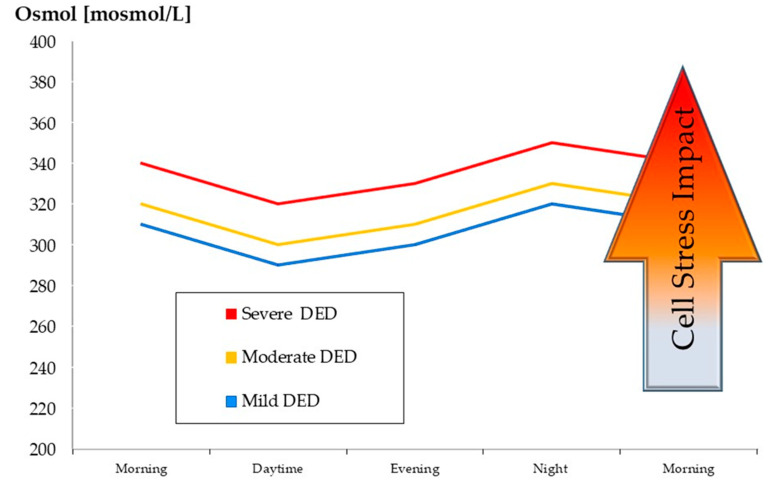
DVO (diurnal variation in osmolarity). Stress amplification by elevated osmolarity level. According to the model of osmokinetics [49,64], it is not only the osmolarity [65] itself and hyperosmolarity that characterize the presence of dry eye, it is more the alteration of osmolarity over a short period of time [49] such as the variation diurnal variation in osmolarity (DVO) together with an elevated level of osmolarity. Desiccation stress caused by the diurnal variation in osmolarity could be amplified by a raised average osmolarity level.

**Figure 2 biology-13-00669-f002:**
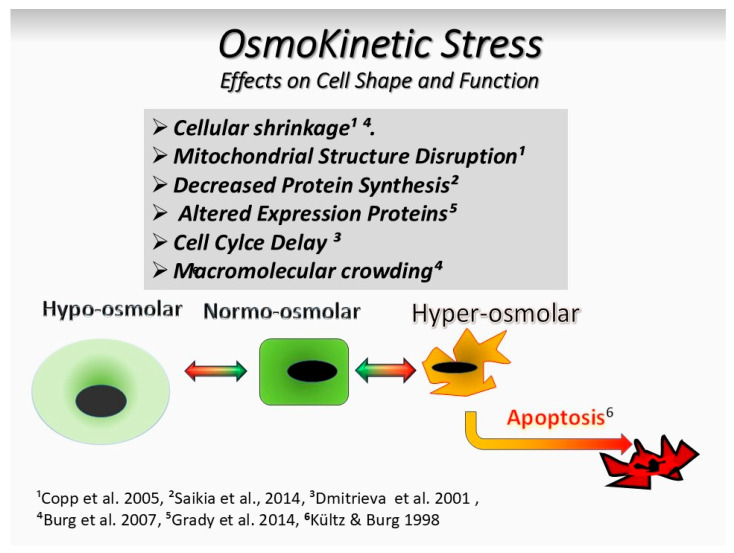
Osmokinetic stress effects on cell shape and function. Osmokinetic stress due to frequent change in osmolarity between normo-osomolar and hyperosmolar conditions could exceed the coping potential of cells and lead to an osmolarity jojo. This could, over time, prevent the ability of osmoadaptation and lead to the activation of pro-inflammatory gene expression and decreased [67], altered, or delayed protein synthesis [67,68]. Eventually, apoptosis is initiated. In this process, the surface of the exposed epithelial cells could shrink [69], shrivel, and develop an abnormal notched surface due to loss of water, i.e., entering a process of crenation.

**Figure 3 biology-13-00669-f003:**
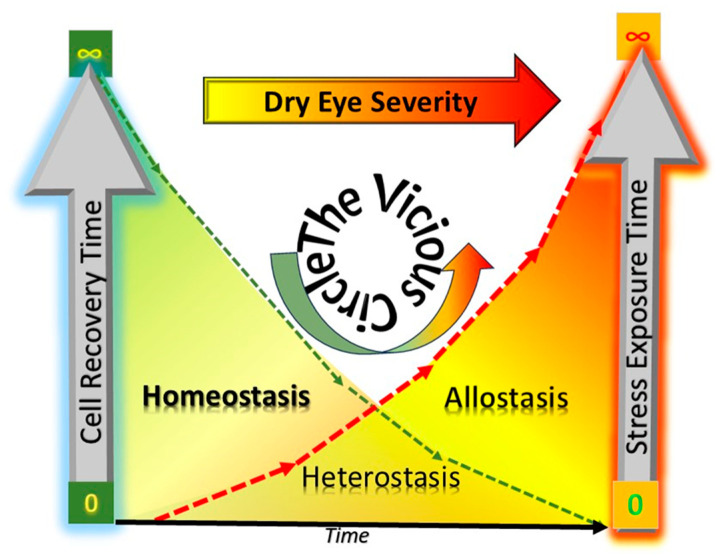
Dry eye severity. Once the vicious circle has been activated and the surface is exposed increasingly to stress, the time for recovery for the cells successively decreases, leading to an increased severity of dry eye symptoms and signs. The autocatalytic character of events with an ever-increasing time of exposure to stress results in a surge of stress fueling the vicious circle [23,24], finally leading to loss of homeostasis and driving the cells at the ocular surface to the stage of allostasis [1].

**Figure 4 biology-13-00669-f004:**
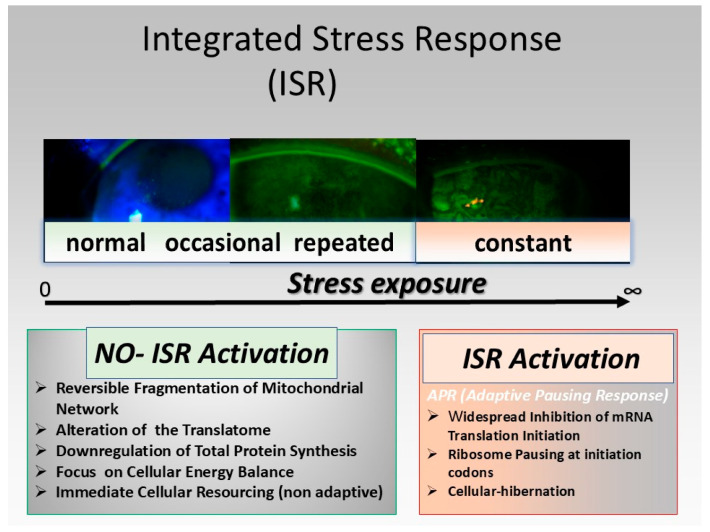
Integrated stress response (ISR). The integrated stress response (ISR) plays a decisive role in the development of dry eye disease as a result of stress exposure. Apparently, the ISR activation entails an enhanced risk for cellular hibernation as well as ribosome pausing, severely altering the cellular response options.

**Figure 5 biology-13-00669-f005:**
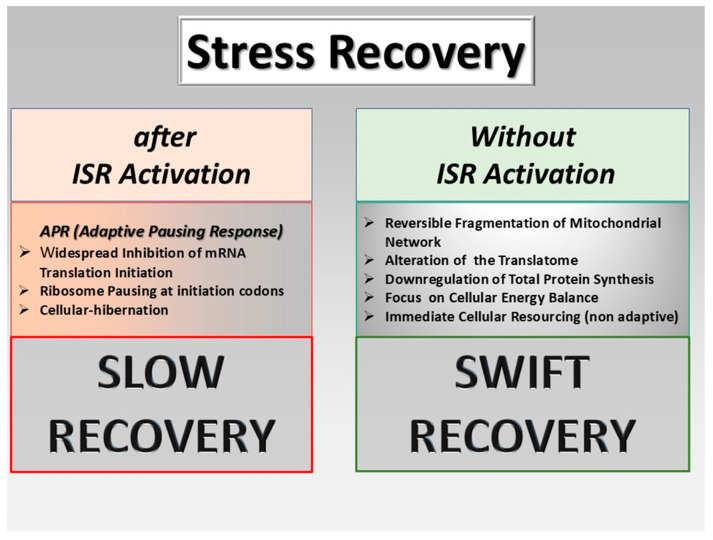
Stress recovery readaption as alternate stress. The ISR (integrated stress response) activation has a decisive role for the options and velocity of recovery from stress.

## Data Availability

Not applicable.

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
