# Peer review of "Cellular Stress in Dry Eye Disease—Key Hub of the Vicious Circle"

_biology, 2024, doi:10.3390/biology13090669_

Round 1

Reviewer 1 Report

Comments and Suggestions for Authors

The Author reviews in this manuscript the several stressors which can impact the ocular surface, taking also in consideration the stress time as a focus point leading to amplification. Several concurrent stressors and estende time can culminate leading to an exhaustion of the normal capacity of the system to act in hormesis. Reaching the limits of stress tolerance leads to a vicious circle which characterize to a disease we refer to as dry eye. To delay the onset of dry eye,  the amelioration of stress factors (or avoidance) is one option. The manuscript is well organized, with explicative figures, and the essential references. The text is biology-oriented, and can be a source of translational information for the readers of the journal.

Author Response

The author would like to thank the reviewer for his positive and encouraging comments. In the re-edited version some more references have been added and the figures have been altered in order to improve their message.  Missing texts to the figures has been added.

Thank you for your time and commitment

Reviewer 2 Report

Comments and Suggestions for Authors

This manuscript provides a comprehensive overview of cellular stress in dry eye disease.  The manuscript could be improved by addressing the following issues:

1.     The introduction should provide more background information on the definition, prevalence, and impact of dry eye disease, as well as the current challenges and gaps in the diagnosis and treatment of this condition.

2.     Page 2, lines 89-91, please add references.

3.     Please add captions to Figures 1 and 2.

4.     Some of the text in Figures 1, 2, 3, 5, 7, and 8 appears to be too small and difficult to read. Please enlarge it for better readability.

5.     Part 2: Cellular stress definition and effects should be better organized, using headings and subheadings to guide the reader.

6.     The paper should also use transitions and connectors to link the different sections and subsections and to show the flow of the argument.

7.     This author should avoid repetition and redundancy in its content. For example, the manuscript mentions some of the same information in different sections, such as the role of hyperosmolarity, inflammation, and apoptosis in dry eye disease. The author should synthesize and summarize the main points and avoid unnecessary details or tangential points.

8.     The author should provide a “conclusion and outlook section” that summarizes the key points of the manuscript and emphasizes the prospects of dry eye disease.

Comments on the Quality of English Language

The manuscript should enhance its writing and structural organization. Additionally, certain figures require revisions for improved clarity.

Author Response

The author would like to thank the reviewer for his positive and encouraging comments signifantly  contributing to improve the manuscript.

According to the suggestions made:

1.)   Introduction was changed according to the suggestions by the reviewer.

2.)   Page 2, lines 89-91,   references  added

3.)     Captions to Figures 1 and 2. Added

4.)     All figures have been redone and improved 

5.)     Structure of text has been altered and rewritten in part  making it easier for the reader.

6.)  Text has been revised and reworded in order to improve the flow and ease of understanding.

7.)    Text  has been changed in order to follow the suggestions

8.)  A chapter “conclusion and outlook section”    summarizing the key points of the manuscript  providing an outlook  has been added.

Thank you for your time and commitment.

Reviewer 3 Report

Comments and Suggestions for Authors

In this manuscript, the author has conducted a comprehensive literature review on the highly interesting topic of oxidative stress in dry eye disease. This topic is not only of great significance for the understanding and treatment of dry eye but also promises to elucidate the mechanisms underlying other conditions on the ocular surface, such as corneal ectasias.

While the scientific content of this review could be relevant, the writing style is really dense, making it difficult to understand. For a more reader-friendly experience and in order to make the manuscript more engaging, it is recommended to do a deep review of its word choice and overall style. Here are some examples about several instances of misspellings, dense language, and the weird format of this manuscript:

- Unclear and repetitive word usage (line 44 "resp", lines 48-49 "elimination or elimination" …).

- Unintroduced abbreviations (line 57 "ER"…).

- Complex and dense sentence structure (lines 244-245-246-247 “An intriguing option is to enhance epithelial cell viability under challenging condition by the application of specific substances alleviating in the endoplasmic reticulum stress such as shown recently for tauroursodeoxycholic acid (TUDCA), a chemical chaperone known to protect against endoplasmic reticulum stress.”).

It is necessary: (1) clarify the terminology; (2) define abbreviations upon first use; (3) address missing punctuation by using proper sentence structure; and (4) ensure clarity, conciseness, and readability.

 It is important to improve the appearance of the provided Figures:

- The Figures looks like PowerPoint slides rather than scientific figures. Remove image backgrounds: replace them with a simple white background to achieve a more professional look consistent with a scientific article.

- The Figures do not add significant value to the written text. Enrich the content: supplement the figures with relevant information to provide more meaningful explanations.

The manuscript does not follow a proper format:

- Excessive spacing between words in multiple sentences (in the majority of the lines throughout the text).

- Lack of consistency in the citation of bibliographic references: sometimes the number is attached to the text and other times it has separation (example: line 54 "stress[6]" and line 47 "mucins [5]").

- Double punctuation at the end of sentences (example: line 56 "directly to the cell..").

- Inconsistent style in the naming of Figures (example: line 90 "Fig-1" and line 97 "Fig.2").

- Figures 1 and 2 do not have captions.

As a result, despite the interesting topic and the veracity of the information provided, there is clear evidence that the work lacks the polish and rigor necessary for publication in a high-impact journal such as Biology. The manuscript should benefit from a thorough revision of the aforementioned terms.

Comments on the Quality of English Language

In order to make the manuscript understandable, a thorough revision of the English language is required.

Author Response

Dear  reviewer,

 Thank you for your valuable time and suggestions.

We have seen over the entire  manuscript as suggested.  We have followed the advice of  improved choice of words and simpler overall style. The entire manuscript has been worked through in order to improve the language and facilitate the understanding. Greatest care was applied to  avoid repetition and redundancy in the text.  This in the intent to ensure clarity, conciseness, and readability.  In detail, the following changes were performed:

  • Unclear and repetitive word usage has been addressed and f
  • All abbreviations have been introduced properly where they occurred first in the text.
  • Missing punctuation, shorter sentences and more proper sentence structure was applied.
  • Considering excessive spacing between words in multiple sentences was checked and corrected as possible - but many spacings issues occur when text is optimized to the margins automatically  by the system. This will be optimized and corrected  by the editor in the final  
  • The consistency in the citation of bibliographic references was addressed throughout the  
  • Double punctuation was identified and deleted.
  • All figures have been redone and improved, the number of figures is reduced.
  • All figures have now consistent style in the naming.
  • All figures have now proper captions.

The author would like to thank the reviewer for his positive and very constructive comments.

Round 2

Reviewer 2 Report

Comments and Suggestions for Authors

The revised manuscript looks good. 

Author Response

Thank you for your reviewing

Reviewer 3 Report

Comments and Suggestions for Authors

While I acknowledge the author' work in revising the manuscript, there are still significant issues that need improvement to bring the manuscript to a publishable standard.

In this second version of the manuscript, the author have improved the text, however, an outstanding effort is still needed to enhance reader comprehension and strengthen the overall impact of the review. It is necessary to provide readers with a more fluent writing style and an easier understanding of the concepts. Consider using professional editing services to improve the overall clarity and readability of the manuscript, as well as informatic writing-assistant tools (such as Grammarly) dedicated to providing guidance and advice in these terms.

In addition, the figures have not been modified to the required standards. The figures are not well-designed and do not effectively convey the information. The author should check other review articles for inspiration on the design and type of figures that usually accompany these manuscripts.

For your reference and guidance, I am enclosing a couple of review articles recently published in the Journal Biology:

(1) “Intestinal Barrier Dysfunction and Gut Microbiota in Non-Alcoholic Fatty Liver Disease: Assessment, Mechanisms, and Therapeutic Considerations” by Changrui Long,Xiaoyan Zhou,Fan Xia andBenjie Zhou. Biology 2024, 13(4), 243; https://doi.org/10.3390/biology13040243 - 06 Apr 2024.

(2) “Overnutrition and Lipotoxicity: Impaired Efferocytosis and Chronic Inflammation as Precursors to Multifaceted Disease Pathogenesis” by Vivek Mann, Alamelu Sundaresan andShishir Shishodia. Biology 2024, 13(4), 241; https://doi.org/10.3390/biology13040241 - 06 Apr 2024.

Comments on the Quality of English Language

Extensive editing required.

Author Response

Thank you for your review.

I have revised the language/sentences of the whole paper and also reduced the number of figures from 8 to 5